HierbaNetV1: a novel feature extraction framework for deep learning-based weed identification

http://orcid.org/0000-0001-8072-3230 Michael Justina 1
Manivasagam Thenmozhi 2 thenmozm@srmist.edu.in
1 Department of Computer Science and Engineering, SRM Institute of Science and Technology , Kattankulathur, Chennai, Tamil Nadu , India
2 Department of Networking and Communications, SRM Institute of Science and Technology , Kattankulathur, Chennai, Tamil Nadu , India
Cirillo Stefano
Electronic publication date: 2024 Nov 22
Publication date: 2024
Volume: 10
Electronic Location ID: e2518
Received 2024 Jun 11; Accepted 2024 Oct 24
Copyright: © 2024 Michael and Manivasagam
Copyright year: 2024
Copyright holder: Michael and Manivasagam
License: This is an open access article distributed under the terms of the Creative Commons Attribution License, which permits unrestricted use, distribution, reproduction and adaptation in any medium and for any purpose provided that it is properly attributed. For attribution, the original author(s), title, publication source (PeerJ Computer Science) and either DOI or URL of the article must be cited.
License URL: https://creativecommons.org/licenses/by/4.0/

Keywords: HierbaNetV1, SorghumWeedDataset_Classification, Convolutional Neural Network (CNN), Classification, Crop-weed identification, Precision agriculture, Feature extraction, Low level feature extraction, High level feature extraction, Feature integration

Funding: The authors received no funding for this work.

==============================
Extracting the essential features and learning the appropriate patterns are the two core character traits of a convolution neural network (CNN). Leveraging the two traits, this research proposes a novel feature extraction framework code-named ‘HierbaNetV1’ that retrieves and learns effective features from an input image. Originality is brought by addressing the problem of varying-sized region of interest (ROI) in an image by extracting features using diversified filters. For every input sample, 3,872 feature maps are generated with multiple levels of complexity. The proposed method integrates low-level and high-level features thus allowing the model to learn intensive and diversified features. As a follow-up of this research, a crop-weed research dataset termed ‘SorghumWeedDataset_Classification’ is acquired and created. This dataset is tested on HierbaNetV1 which is compared against pre-trained models and state-of-the-art (SOTA) architectures. Experimental results show HierbaNetV1 outperforms other architectures with an accuracy of 98.06%. An ablation study and component analysis are conducted to demonstrate the effectiveness of HierbaNetV1. Validated against benchmark weed datasets, the study also exhibits that our suggested approach performs well in terms of generalization across a wide variety of crops and weeds. To facilitate further research, HierbaNetV1 weights and implementation are made accessible to the research community on GitHub. To extend the research to practicality, the proposed method is incorporated with a real-time application named HierbaApp that assists farmers in differentiating crops from weeds. Future enhancements for this research are outlined in this article and are currently underway.

Introduction

Feature extraction and pattern recognition are the salient characteristics of convolutional neural networks (CNN or ConvNet). The effectiveness of a feature extraction technique differs from one architecture to another. Conventional classification algorithms such as Visual Geometry Group (VGGs) extract generalized features from deep networks (Simonyan & Zisserman, 2014), inceptions extract multi-scale features from deep and wide networks (Szegedy et al., 2015), Residual networks (ResNets) extract features from skip connection networks (He et al., 2016), densely connected convolutional networks (DenseNets) extract discriminative features from deep networks with feature reuse capability (Huang et al., 2017), and MobileNets extract features from light-weight networks suitable for all applications (Howard et al., 2017).

HierbaNetV1, our novel feature extraction technique is intended to extract intensive and diverse features at multiple levels of complexity, using varying-sized kernels. Unlike traditional architectures, it maintains high effectiveness across inputs, regardless of the Region of Interest (ROI) size. By balancing both depth and width in the network, HierbaNetV1 avoids the drawbacks of overly deep networks. The architecture enables feature propagation thus avoiding the problem of vanishing gradient. Additionally, the use of the LeakyReLU activation function (Dubey & Jain, 2019) addresses the common “dying ReLU” issue seen in other models, ensuring robust pattern recognition across varying input complexities.

CNN through its feature extraction demonstrates promising results in various aspects of crop-weed management including crop-weed identification (Calderara-Cea et al., 2024), crop-weed detection (Asad, Anwar & Bais, 2023), weed mapping (Wang et al., 2023), and weed segmentation (Celikkan et al., 2023). CNN-based weed architectures and networks (Xu et al., 2023) are specifically designed to differentiate weeds from crops, facilitating autonomous weed removal for agriculturalists. Weeds must be effectively managed as they contribute significantly to crop yield loss, with studies indicating that weeds account for approximately 45% of such losses, followed by insects (30%), diseases (20%), and other factors (5%). Crop yield losses attributed to weeds can range from a minimum of 15% to as high as 76% (Gharde et al., 2018). Beyond impacting productivity, weeds also lead to substantial economic losses for farmers.

Motivation and significance

Artificial intelligence (AI) is making significant strides across various sectors; however, its application in agriculture, particularly in countries like India, is still in its early stages. A significant challenge in farming is the labor-intensive process of weeding, especially since early-stage weeds often closely resemble crops. Although AI-driven weed removal techniques have demonstrated considerable promise in tackling this issue, their practical adoption on a wider scale is still limited.

This research aims to support weed scientists and the agricultural community by developing a robust AI-based weed identification system for the precise early detection of weeds. Furthermore, this research enables real-time weed detection by incorporating the proposed model into an Android mobile application capable of distinguishing crops from weeds. By reducing or eliminating weed growth, this system enhances crop yield and supports healthier plant growth, ultimately contributing to the economic growth of the country.

Contributions

The contribution of the article is summarized as follows: 1. Novel framework: This research designs and develops a novel feature extraction framework code named HierbaNetV1, that retrieves and learns effective features from an input image.

2. Own dataset: To encourage weed research, a crop-weed dataset termed SorghumWeedDataset_Classification is created and is publicly available in the Mendeley Data repository at https://data.mendeley.com/datasets/4gkcyxjyss/1. To the best of our knowledge, the above-mentioned dataset is the first open-access crop-weed research dataset from the Indian field to deal with crop-weed classification.

3. Releases model weights: Trained model weights of HierbaNetV1 are publicly available to the research community to encourage further extensions in the architecture. It is available in GitHub at https://github.com/JustinaMichael/HierbaNetV1-A-Novel-CNN-Architecture in Hierarchical Data Format version 5 (HDF5) data file format.

4. Releases source code: The proposed architecture is implemented in Python and the code is released on the reproducibility platform CodeOcean at https://codeocean.com/capsule/5579071/tree/v1.

5. Real-time application: To bring weed research into practicality, this research employs the proposed HierbaNetV1 with a specially designed user-friendly Android mobile application named HierbaApp leveraging CNNs for on-the-spot weed identification by farmers. The application can be accessible in the Google Play Store at https://play.google.com/store/apps/details?id=com.hierba.app

The rest of the article is organized as follows: ‘Related Works’ presents a summary of current weed detection architectures, outlining their methodologies and highlighting the research gaps addressed by these existing approaches. It also discusses the problem addressed in this research. In ‘Materials and Methods’ data acquisition, dataset creation, data pre-processing pipeline, and the novel feature extraction technique HierbaNetV1 are discussed. ‘Implementation of HIERBANETV1’ explains the implementation of HierbaNetV1 with a sequential algorithm, model parameters, and tuned hyper-parameters. The generated feature maps are portrayed for better understanding. ‘Results and Discussions’ depicts and discusses the model performance through various result analysis, ablation studies, component analysis, and real-time inference. Eventually, future improvements are explored along with the conclusion in ‘Conclusions and Future Enhancements’.

Related works

Recent studies have demonstrated the effectiveness of CNNs, contributing to advancements in various aspects of crop-weed classification. This section reviews the state-of-the-art (SOTA) architectures for crop-weed identification, with a summary of key models presented in Table 1.

Table 1 State-Of-The-Art methodologies for crop-weed identification.

Reference	Dataset (D), methodology (M) and research gap addressed (RGA)	Task and result	
Xu, Jin & Guo (2024)	D: Cotton seedlings dataset M: Cotton seedling identification model RGA: Accuracy and timeliness	Segmentation with 95.75% ACCa	
Mckay et al. (2024)	D: T1_miling, T2_miling, YC datasets M: U-Net with ResNet18, ResNet34, VGG16 RGA: Focuses on crop than weeds	Semantic segmentation with 0.84 APa	
Jiang et al. (2024)	D: Cropandweed, Sugar Beet 2020 datasets M: SWFormer RGA: Class imbalance	Semantic segmentation with mAPa of 76.54% and 61.24%	
Naik & Chaubey (2024)	D: Crop and weed detection and Plant Seedling datasets M: Region-Based Convolutional Neural Networks RGA: Manual detection and categorization of weeds	Classification with an ACCa of 95.88% and 97.29%	
Thiagarajan, Vijayalakshmi & Grace (2024)	D: Beans dataset M: SegNet, ResUNet, UNet RGA: Hyperparameter optimization in focal loss	Semantic segmentation with 0.8444 IOUa	
Modi et al. (2023)	D: Sugarcane dataset; M: DarkNet53 RGA: High-cost weed detection systems	Classification with 96.6% ACCa	
Sahin et al. (2023)	D: Sunflower dataset M: Conditional Random Field with U-Net RGA: Class imbalance (Few positive class)	Semantic segmentation with a maximum IOUa of 0.990	
Jiang, Afzaal & Lee (2022)	D: 1006 weed dataset M: Swin Transformer, SegFormer, Segmenter RGA: Localization and classification	Semantic segmentation with mAcca of 75.18%	
Zhang (2023)	D: Deepweeds dataset M: Hybrid CNN-Transformer model RGA: Network computation and time efficiency	Classification with 96.08% ACCa	
Su et al. (2021)	D: Narrabri and Bonn dataset M: Random Image Cropping And Patching (RICAP) RGA: Data augmentation for semantic segmentation	Semantic segmentation with mAcca of 94.02% and 98.51%	
Note:

a Metrics: ACC (Accuracy), AP (Average precision), mAcc (Mean Accuracy), mAP (Mean Average Precision), IOU (Intersection Over Union score).

Recent research has shifted focus toward crop identification rather than focusing on several categories of weeds, thus introducing a novel approach for weed detection (Mckay et al., 2024). The performance of a CNN model is primarily based on its data. Therefore, studies increasingly emphasize data augmentation techniques such as random image cropping and patching (RICAP) to improve model robustness under varying lighting and environmental conditions (Su et al., 2021; Fawakherji et al., 2024). A balanced dataset prevents the model from becoming biased towards one class. To address this, researchers have employed advanced methods like SWFormer and Conditional Random Fields (Jiang et al., 2024; Sahin et al., 2023). When dealing with small and complex datasets, traditional CNNs often struggle to generalize effectively. To address this, research highlights the integration of CNNs with other machine learning techniques to improve the classification of crop-weed features, while also enhancing model interpretability (Urmashev et al., 2021; Tao & Wei, 2022).

Alongside, transfer learning with pre-trained CNNs has proven effective in developing multi-class models that successfully classify various crop and weed species, addressing the challenge of biodiversity in agricultural settings (Li et al., 2024; Gao et al., 2024). In parallel, significant work has been devoted to hyper-parameter optimization, recognizing its critical role in controlling the performance of CNN-based models for crop-weed classification (Thiagarajan, Vijayalakshmi & Grace, 2024; Ajayi, Ibrahim & Adegboyega, 2024).

Further research has focused on weed detection and segmentation to achieve precise weeding through improved object localization. In these techniques, CNN serves as the backbone feature extractor for models such as region-based CNN (R-CNN), Fast R-CNN, Faster R-CNN, Mask R-CNN (MRCNN), and YOLO (Naik & Chaubey, 2024; Zhang et al., 2023; Zheng et al., 2024). Also, hybrid models combining CNNs with transformer architectures, such as Swin Transformer, SegFormer, Segmenter, and other hybrid CNN-Transformer models, have been introduced to enhance computational efficiency and speed in complex agricultural scenarios (Zhang, 2023; Jiang, Afzaal & Lee, 2022; Kala et al., 2024).

Research has also focused on real-time weed detection in field environments, achieving improvements in accuracy, processing speed, and time efficiency, with direct applications in agricultural practices (Xu, Jin & Guo, 2024; Naik & Chaubey, 2024; Goyal, Nath & Niranjan, 2024). Furthermore, CNN integration with drone technology has enabled aerial surveillance for real-time weed detection across expansive agricultural fields (Mesías-Ruiz et al., 2024; Seiche, Wittstruck & Jarmer, 2024). Comprehensive reviews on crop-weed identification have further synthesized findings from various studies, underscoring CNNs’ potential to revolutionize precision agriculture by enabling effective weed management (Qu & Su, 2024; Adhinata, Wahyono & Sumiharto, 2024; Hu et al., 2024).

Highlights and limitations of prior work: Research in the field of weed identification evolves with greater advancements and has shown significant contributions as observed from the literature. This survey highlights that architectures developed for crop-weed classification and segmentation perform tremendous tasks resulting in high performance. Weed research is advanced as contemporary studies focus on precisely locating the ROIs with considerably high accuracies. As a result, systems become more scalable and efficient. However, limitations arise in feature extraction methods when they fail to precisely identify different-sized ROIs. Smaller ROIs are more challenging to identify accurately and call for a sophisticated feature extraction strategy that is lacking from the existing approaches. There are more contributions in the technical environment, but fewer in the actual application of these promising discoveries. By using the same in real fields, it may be possible to enable autonomous weeding, which will enhance crop and soil health and result in higher crop yields with lower costs and labor optimization.

Research gaps and challenges addressed: The focus of current feature extraction techniques is on a particular level of complexity, which makes it difficult to recognize ROIs of different sizes. The proposed method fills the research gap by addressing the problem of varying-sized ROIs by introducing four diversified filters that generate a rich set of features. This makes it possible to recognize weeds and crops in both early and later stages thus facilitating early weed removal. Additionally, our method places a strong emphasis on low-level feature propagation to avoid the vanishing gradient problem. This enables deep learning of the basic features, which is crucial to distinguish one class from the other. Furthermore, the model overcomes the problem of over-fitting by adding a drop-out layer and integrates early stopping to address the Bias-Variance trade-off issue.

Materials and Methods

Dataset: from acquisition to pre-processing

As a part of this research, crop-weed images are acquired. The research objects focused on data acquisition are the early growth stages of sorghum (Class_0), grass (Class_1), and broadleaf weed (Class_2) which are depicted in Fig. 1. The data is acquired during April and May 2023 from Sri Ramaswamy Memorial (SRM) Care Farm, Chengalpattu district, Tamil Nadu, India. Data is captured in the form of red, green, and blue (RGB) images using Canon Electro-Optical system (EOS) 80 D–a digital single lens reflex (DSLR) camera with a sensor type of 22.3 mm × 14.9 mm Complementary Metal Oxide Semiconductor (CMOS). Three different weather conditions such as sun, strong wind, and light rain are used to collect data samples in the morning and afternoon light conditions. The entire data acquisition process and detailed description of the datasets are briefed in the data article (Michael & Manivasagam, 2023c).

Figure 1 Sample research objects from SorghumWeedDataset_Classification: (A–C) Sorghum samples, (D–F) grass weed samples, and (G–I) Broadleaf weed samples.

SorghumWeedDataset_Classification, a crop-weed research dataset is framed from the acquired images and is publicly available in the Mendeley Data repository (Michael & Manivasagam, 2023b). This dataset solves the crop-weed classification problem in smart weeding. The SorghumWeedDataset_Classification dataset contains 4,312 data samples, which are used for crop-weed image classification in deep learning. To ease the classification task, the dataset is split into Train: Validate: Test (TVT) with a ratio of 7:2:1 which is detailed in Table 2. All data samples are in RGB and JPG format.

Table 2 SorghumWeedDataset_Classification TVT split.

Class ID	Class name	Train (70%)	Val (20%)	Test (10%)	Total (100%)	
Class 0	Sorghum	983	281	140	1,404	
Class 1	Grass	1,027	293	147	1,467	
Class 2	Broadleaf weed	1,009	288	144	1,441	
Total	3,019	862	431	4,312	

In the preprocessing pipeline, the samples undergo resizing, augmentation, and normalization through the following steps; The original size of an acquired data sample is 6,000×4,000 pixels. HierbaNetV1 accepts an input image of size 224×224×3 to reduce the computational complexity. Hence, all classification data samples are re-sized to 224×224 pixels without information loss. During the learning process, the training samples are augmented with 45-degree rotation, 25% zoom, 25% slant, horizontal and vertical flip, 30% width and height shift, and brightness adjustment in the range of 0.2 to 0.9 to avoid the problem of under-fitting. The parameter values are assigned to every augmentation method after carrying out an experimental study that generates effective variants of an original sample (Michael & Manivasagam, 2023a). The data samples are normalized between 0 and 1 with a factor of 1/255 to handle the complexity.

Feature extraction with HierbaNetV1

Purpose of invention: “Hierba” is a Spanish word that means “weed”–a plant that grows in an unwanted place. HierbaNetV1 is thus named because the primary reason for building this architecture is to extract effective crop and weed features and thus address the crop-weed identification problem in precision agriculture. However, HierbaNetV1 extracts intensive features in other computer vision projects as well.

Ideation behind HierbaNetV1: The idea behind HierbaNetV1 is to perform intensive feature extraction from each data sample focusing on multiple levels of complexity, irrespective of the ROI size, and emphasize low-level features consistently. In contrast to the majority of current methods, HierbaNetV1 employs diversified filters to produce rich feature maps while concentrating on lowering the number of convolutional layers to simplify the model with fewer parameters. This novel architecture is patented at the Indian Patent Office (Patent application no. 202441050194) (Michael & Manivasagam, 2024a). The workflow of the proposed architecture is briefed in the research video article (Michael & Manivasagam, 2024c).

Principle of HierbaNetV1: The novelty and key characteristic of HierbaNetV1 is its three-step intensive feature extraction technique. Firstly, the four diversified filters with kernel sizes 1×1, 3×3, 5×5, and 7×7 extract distinct features to form a set of high-level features ( FHL), which is denoted in Eq. (1).

(1) FHL={FHL1,FHL2,FHL3,....FHLn}

Secondly, the conventional filters with kernel size 3×3 extract basic features to form a set of low-level features ( FLL) denotes Eq. (2).

(2) FLL={FLL1,FLL2,FLL3,....FLLn}

Thirdly, FHL and FLL integrate to form a union of high-level and low-level features, described as F in Eq. (3).

(3) F={FHL+FLL}

F generated at Block I is denoted as FB1, and the one from Block II is denoted as FB2. FB1 and FB2 are F variants formed at two intermediate input scales.

Design framework of HierbaNetV1

HierbaNetV1 receives an image of size 224×224×3 and predicts the respective class as the output. Figure 2 depicts the base architecture of HierbaNetV1 with abstracted connecting blocks. The connecting blocks HierbaNetV1_BLOCK I and HierbaNetV1_BLOCK II are illustrated elaborately in Figs. 3 and 4 respectively.

Figure 2 Architecture of HierbaNetV1.

Our model, firstly resizes each input image to 224 × 224 × 3, convolves, batch-normalizes, and activates neurons with LeakyReLU followed by downsampling. Secondly, runs a two-block feature extraction technique with two modules. Module I extracts high-level features ( FHL) using four diversified filters of multiple levels of complexity. Module II extracts low-level features ( FHL). Progressively, feature set F is formed by integrating FHL and FLL, with which the model is trained. Thirdly, the generated features are flattened, and 20% of its neurons are dropped followed by three-way softmax activation. Lastly, the trained model predicts the class of the input sample.

Figure 3 Architecture of HierbaNetV1_Block I: Block I and Block II define the novelty and key characteristics of our proposed model.

Each block follows three crucial steps: Firstly, the four diversified filters with kernel sizes 1×1, 3×3, 5×5, and 7×7 extract high-level features (FHL). Secondly, low-level features (FLL) are extracted by filters of kernel sizes 3×3. Thirdly, FHL and FLL are integrated to form F thus producing a rich set of features with multiple levels of complexity.

Figure 4 Architecture of HierbaNetV1_Block II: Block II extracts the features similar to Block I, but with downsampled input samples.

HierbaNetV1_Base: As the first step, the model receives one input sample from the training set and splits it into three channels (red, green, and blue) as described in Eq. (4).

(4) s={sr,sg,sb}

The input sample convolves with the kernel (LeCun & Bengio, 1995), batch-normalizes the inter-layer’s output (Ioffe & Szegedy, 2015), and activates neurons using LeakyReLU (Xu et al., 2020), followed by downsampling with Max-Pooling (Riesenhuber & Poggio, 1999). Equation (5) defines the relation between the input sample s, its respective filter f, and the convolved output sc.

(5) sc=s∗f

The convolved output sc is batch normalized with batch mean bm, batch variance bv2, scale parameter p1, shift parameter p2, and c=0.001 a constant to produce a batch normalized output sbn defined in Eq. (6).

(6) sbn=p1×sc−bmbv2+c+p2

The batch normalized values are activated for all positive values accordingly, whereas the negative values are multiplied with 0.001 before activation, thus producing f(sbn) defined in Eq. (7).

(7) f(sbn)={sbn,ifsbn>0.0.001sbn,otherwise.

Equation (8) reduces the dimension of feature maps using Max-Pooling to minimize the computational complexity and also to produce deep features. M, N, and Ch represent the width, height, and channels respectively. Fmp denotes the downsampled feature maps, which are passed as input to the first connecting block, HierbaNet_Block I.

(8) Fmp∈FM2×N2×Ch

HierbaNetV1_Block I: Feature maps are fed to HierbaNetV1_Block I from HierbaNetV1_Base. New features are extracted parallelly in two modules as illustrated in Fig. 3. Module I extracts high-level features using four diversified filters with kernel sizes 1×1, 3×3, 5×5, and 7×7 as described in Eq. (9) and integrates the features as a single unit. Sequentially, the combined features convolve with the kernel, batch-normalize, and activate neurons thrice. This workflow is exhibited at the left hand side (LHS) of Fig. 3.

(9) FHL∈{X1×1×3,X3×3×3,X5×5×3,X7×7×3}

Module II extracts low-level features using the conventional filter with kernel size 3×3 which is portrayed at the right hand side (RHS) of Fig. 3 and described in Eq. (10).

(10) FLL∈X3×3×3

Concatenated feature maps from Module I and Module II are given as input to Base_Conv2. Progressively, HierbaNetV1_Base convolves, batch-normalizes, and activates neurons using LeakyReLU, followed by downsampling as pictured in Fig. 2.

HierbaNetV1_Block II: Feature maps from Base_MaxPool2 are fed to HierbaNetV1_Block II which performs similar operations as HierbaNetV1_Block I, but with down-sampled feature maps. Figure 4 portrays the particulars of HierbaNetV1_Block II with respective dimensions.

Equations (11) and (12) represent that each block generates 64 low-level feature maps and 1,280 high-level feature maps. A total sum of 3,872 feature maps is generated by HierbaNetV1 for a sample.

(11) FLL={F1,F2,F3,....,F64}

(12) FHL={F1,F2,F3,....,F1280}

HierbaNetV1_Base: Block II outputs 576 feature maps with a 7×7 dimension each. Global average pooling flattens these multi-dimensional features into one-dimensional features by extracting one feature from each feature map, thus making 576 features. Neurons are dropped with a probability of 0.2 to avoid over-fitting (Kromer-Edwards, Castanheira & Oliveira, 2023). Three-way softmax in the dense layer distributes the output among the three class labels (Bridle, 1990). Ultimately, the model predicts the input class of the given input sample.

Implementation of hierbanetv1

Training platform

The development of HierbaNetV1 was carried out using the following hardware and software configurations. Google Colaboratory (Bisong, 2019), an online cloud platform with a Colab Pro subscription, is used for training the models. The runtime allotted an NVIDIA-SMI Driver version 525.85.12 with CUDA Toolkit version 12.0 and 89.6 gigabytes of high RAM with NVIDIA A100. The model training took 2.00 h, consuming 45 compute units to complete the training, validation, and testing process. Model training, testing, and result visualization are done in Python 3.10.12. Libraries that supported the implementation are keras 2.12.0 for deep learning, tensorflow 2.12.0 for creating machine learning frameworks, scikit-learn 1.2.2 for machine learning, pillow 9.4.0 for Imaging, numpy 1.22.4 for array computing, matplotlib 3.7.1 for Python plotting package, seaborn 0.12.2 for statistical data visualization, and pandas 1.5.3 for data analysis.

Sequential algorithm design

The comprehensive pseudocode of the layers in HierbaNetV1 and its workflow is briefed in Algorithm 1, which is related to Fig. 2. User-defined Feature_Integration method extracts and integrates the low-level and high-level features, which are detailed in Algorithm 2, which is related to Figs. 3 and 4. In this, Module I-FeatureExtraction_HighLevel generates the high-level features, Module II-FeatureExtraction_LowLevel generates the low-level features, and FeatureIntegration integrates the features generated from Module I and Module II. The details of 19 convolutional layers along with the growth rate of feature maps are illustrated in Table 3. HierbaNetV1 has 72 layers in total. Model hyper-parameters that are fine-tuned after several experiments are tabulated in Table 4 (Kingma & Ba, 2014; Srivastava et al., 2014).

Algorithm 1 Pseudocode for HierbaNetV1

 function HierbaNetV1(Instance of an input image)	
   input_image = inputImage(224, 224, 3)	
   model = conv_Bn_Act(input_image, 32, 3, 3)	
   model = MaxPooling((2, 2), (2, 2), “same”)(model) ▹ HierbaNetV1_Block I	
   model = feature_Integration(model)	
   model = conv_Bn_Act(model, 576, 3, 3)	
   model = MaxPooling((2, 2), (2, 2), “same”)(model) ▹ HierbaNetV1_Block II	
   model = feature_Integration(model)	
   model = conv_Bn_Act(model, 576, 3, 3)	
   model = MaxPooling((2, 2), (2, 2), “same”)(model)	
   model = GlobalAveragePooling(“channels_last”)(model)	
   model = Dropout(0.2, None)(model)	
   model = Dense(number_of_classes, “softmax”, “glorot_uniform”, “zeros”)(model)	
   HierbaNetV1 = Model(input_image, model, “HierbaNetV1”)	
 end function	

Algorithm 2 Pseudocode for feature integration

 function Extract_Integrate(output_high, output_low, output)	
    highlevel_features = featureExtraction_HighLevel(output)	
    lowlevel_features = featureExtraction_LowLevel(output)	
    output = Concatenate([highlevel_features, lowlevel_features], axis = 3)	
▹High-level and Low-level Feature Concatenation	
   return output	
 end function	
 function FeatureExtraction_LowLevel(output_low)	
    output_low = conv_Bn_Act(output_low, 64, 3, 3)	
    output_low = MaxPooling((2, 2), (2, 2), “same”)(output_low)	
▹Dimensionality Reduction	
    return output_low	
 end function	
 function FeatureExtraction_HighLevel(output_high)	
    kernel_1 × 1 = conv_Bn_Act(output_high, 64, 1, 1)	
    kernel_3 × 3 = conv_Bn_Act(output_high, 64, 3, 3)	
    kernel_5 × 5 = conv_Bn_Act(output_high, 64, 5, 5)	
    kernel_7 × 7 = conv_Bn_Act(output_high, 64, 7, 7)	
    output_high = Concatenate([kernel_1 × 1, kernel_3 × 3, kernel_5 × 5, kernel_7 × 7],	
    axis = 3) ▹Feature Concatenation	
    output_high = conv_Bn_Act(output_high, 256, 3, 3)	
    output_high = conv_Bn_Act(output_high, 256, 3, 3)	
    output_high = conv_Bn_Act(output_high, 512, 3, 3)	
    output_high = MaxPooling((2, 2), (2, 2), “same”)(output_high)	
▹Dimensionality Reduction	
    return output_high	
 end function	

Table 3 HierbaNetV1 model summary of convolutional layers.

Layer#	Layer name	Output shape	Parameters	KernelSize	Filters	Growthrate of featuremaps	
1	Base_Conv1 (Conv2D)	(None, 224, 224, 32)	896	3 × 3	32	32	
5	B1_HL_Conv1 (Conv2D)	(None, 112, 112, 64)	2,112	1 × 1	64	96	
6	B1_HL_Conv2 (Conv2D)	(None, 112, 112, 64)	18,496	3 × 3	64	160	
7	B1_HL_Conv3 (Conv2D)	(None, 112, 112, 64)	51,264	5 × 5	64	224	
8	B1_HL_Conv4 (Conv2D)	(None, 112, 112, 64)	100,416	7 × 7	64	288	
18	B1_HL_Conv5 (Conv2D)	(None, 112, 112, 256)	590,080	3 × 3	256	544	
21	B1_HL_Conv6 (Conv2D)	(None, 112, 112, 256)	590,080	3 × 3	256	800	
24	B1_HL_Conv7 (Conv2D)	(None, 112, 112, 512)	1,180,160	3 × 3	512	1,312	
25	B1_LL_Conv1 (Conv2D)	(None, 112, 112, 64)	18,496	3 × 3	64	1,376	
33	Base_Conv2 (Conv2D)	(None, 56, 56, 576)	2,986,560	3 × 3	576	1,952	
37	B2_HL_Conv1 (Conv2D)	(None, 28, 28, 64)	36,928	1 × 1	64	2,016	
38	B2_HL_Conv2 (Conv2D)	(None, 28, 28, 64)	331,840	3 × 3	64	2,080	
39	B2_HL_Conv3 (Conv2D)	(None, 28, 28, 64)	921,664	5 × 5	64	2,144	
40	B2_HL_Conv4 (Conv2D)	(None, 28, 28, 64)	1,806,400	7 × 7	64	2,208	
50	B2_HL_Conv5 (Conv2D)	(None, 28, 28, 256)	590,080	3 × 3	256	2,464	
53	B2_HL_Conv6 (Conv2D)	(None, 28, 28, 256)	590,080	3 × 3	256	2,720	
56	B2_HL_Conv7 (Conv2D)	(None, 28, 28, 512)	1,180,160	3 × 3	512	3,232	
57	B2_LL_Conv1 (Conv2D)	(None, 28, 28, 64)	331,840	3 × 3	64	3,296	
65	Base_Conv3 (Conv2D)	(None, 14, 14, 576)	2,986,560	3 × 3	576	3,872	
Parametric details of the 72 layers in HierbaNetV1	
Total params: 14,331,331	
Trainable params: 14,323,587	
Non-trainable params: 7,744	
Total feature maps generated: 3,872	

Table 4 Hyperparameter tuning results for HierbaNetV1.

Hyper-parameters	Optimized values	
Input_shape	224, 224, 3	
Optimizer	Adam	
Optimizer-learning_rate	0.001	
Optimizer-epsilon	1e-07	
Convolution-filters	32, 64, 256, 512, and 576	
Convolution-kernel_size	1×1, 3×3, 5×5, and 7×7	
Convolution-padding	Same	
Convolution-strides	1×1	
Dimensionality reduction	MaxPool	
MaxPooling-pool_size	2×2	
MaxPooling-padding	same	
MaxPooling-strides	2×2	
Hidden layer activation	LeakyReLU	
Dropout rate	0.2	
Dense layer activation	SoftMax	
‘k’ in Stratified k-fold cross validation	10	
Epochs in model training	50	
Batch_size in model training	32	

Details of learning

Initial weights of HierbaNetV1 are initialized using Glorot Uniform (Glorot & Bengio, 2010). The model uses stratified 10-fold cross-validation with 50 training epochs for each fold, thus generalizing the model. However, the model calls for early stopping at each fold by monitoring the validation accuracy, with a patience value of five, to regularize the model. The maximum and minimum number of epochs where early stopping is triggered is 10 epochs for fold 1, fold 5, fold 6, and six epochs for fold 4, and fold 8, respectively, thus avoiding over-fitting. Thirty-two training samples are present in a single batch causing 218 iterations to complete one training epoch. Best model weights from each fold are carried over to the subsequent folds to resume training. All these factors contributed to the better convergence of the model.

Stratified 10-fold cross-validation is used for training and validating HierbaNetV1 on the SorghumWeedDataset_Classification. The highest and lowest training accuracy is observed to be 0.9843 and 0.8857 in folds 10 and 1 respectively. Similarly, the highest and lowest training loss is 0.3193 and 0.0442 in folds 1 and 10. The training and validation accuracies and losses for the 10 folds are graphed in Fig. 5. HierbaNetV1 generates 3,872 feature maps from the 19 convolutional layers. One sample feature map from each convolutional layer is illustrated in Fig. 6.

Figure 5 Accuracy and loss graphs of the stratified 10-fold cross-validation of HierbaNetV1 on SorghumWeedDataset_Classification.

(A) Training and validation accuracies, (B) training and validation losses, (C) zoomed-in view of training and validation accuracies, (D) zoomed-in view of training and validation losses.

Figure 6 Illustration of feature maps from the 19 consecutive convolutional layers in HierbaNetV1 on the test image “SorghumTest(1).jpeg” from the SorghumWeedDataset_Classification dataset.

Results and discussions

Extensive experimental research is performed on the proposed architecture and the results prove that HierbaNetV1 has outperformed other techniques.

Model testing

Testing on own dataset

HierbaNetV1 is tested on the 431 test images from SorghumWeedDataset_Classification. It produces an accuracy of 0.9860 and a loss of 0.07. The confusion matrix depicted in Figs. 7A–7C illustrates the confusion matrices for class0, class1, and class2 respectively. HierbaNetV1 has a significant reduction in the false negatives (FN) with a sum of six. It misclassifies two sorghum crops as broadleaf weeds and four grass weeds as sorghum crops. Figures 7D–7F portrays the receiver operating characteristic-area under curve (ROC-AUC) for class0, class1, and class2 as 0.9960, 0.9975, and 1.0 respectively. The ROC-AUC for class 2 is at its maximum as no broadleaf weeds are misclassified for grass or sorghum crops. Figures 7G–7I illustrates the precision recall-area under curve (PR-AUC) for class0, class1, and class2 as 0.9893, 0.9961, and 0.9999 respectively. As can be observed from the confusion matrix there are zero misclassifications in class 2 and two misclassifications in class 2 which is considerably lesser. This results in a greater PR-AUC value for the class (0.9999) than for classes 0 and 1. Results produced by our proposed method are validated manually by agronomists and stated to be accurate. Using stratified 10-fold cross-validation and 3,881 images with a resolution of 224×224, the training and validation of HierbaNetV1 required 2 h. Time complexity analysis shows HierbaNetV1 takes 0.07 ms for testing a 224×224 sample image.

Figure 7 Confusion matrix, ROC curve and PR curve of SorghumWeedDataset_Classification with HierbaNetV1 for individual classes.

(A) Confusion matrix of class0, (B) confusion matrix of class1, (C) confusion matrix of class2, (D) ROC curve of class0, (E) ROC curve of class1, (F) ROC curve of class2, (G) PR curve of class0, (H) PR curve of class1, and (I) PR curve of class2.

Testing on benchmark dataset

To further assess the capability of HierbaNetV1, we have evaluated it against three crop-weed datasets, and the results are compared against existing pre-trained models. This analysis validates its generalizability and robustness across different crop types and environmental conditions.

(a) Soybean weed dataset: The soybean weed dataset (Alessandro et al., 2017) has 15,336 images belonging to four classes such as soil (3,249), soybean (7,376), grass (3,520), and broadleaf weeds (1,191). In this work, we have considered a balanced Soybean weed dataset by choosing the first 1,191 images from each class. HierbaNetV1 is evaluated against other pre-trained models using this dataset. According to the results, among various pre-trained models, our suggested method yields the best accuracy of 98.75%, while ResNet152V2 yields the next highest accuracy of 96.64%. Confusion matrices of the soybean weed dataset using HierbaNetV1 and ResNet152V2 are portrayed in Fig. 8.

Figure 8 Confusion matrix of soybean weeds dataset using (A) HierbaNetV1 and (B) ResNet152V2.

(b) Deepweeds dataset: The Deepweeds dataset has 17,509 images belonging to nine classes (Olsen et al., 2019). Classes from one to eight contain images in the range of 1,009 to 1,125, whereas the ninth is a negative class with 9,106 images. To consider a balanced dataset, the first eight classes are considered ignoring the ninth class. Using this dataset, we compare HierbaNetV1 with other pre-trained models. The two models with the highest accuracy, HierbaNetV1, and DenseNet201, respectively, are 93.99% and 81.84%. Confusion matrices of the Deepweeds dataset using HierbaNetV1 and DesNet201 are illustrated in Fig. 9.

Figure 9 Confusion Matrix of Deepweeds dataset using (A) HierbaNetV1, and (B) DenseNet201.

(c) CottonWeedID15: The CottonWeedID15 dataset (Chen et al., 2022) has 5,187 images belonging to 15 classes. The work considers a balanced dataset with 61 images in each class. This dataset is utilized to assess HierbaNetV1 in comparison to other pre-trained models. With an accuracy of 82.22%, HierbaNetV1 has the best accuracy, followed by InceptionV3 with 77.78%. Confusion matrices of the CottonWeedID15 dataset using HierbaNetV1 and InceptionV3 are depicted in Fig. 10.

Figure 10 Confusion matrix of cotton weeds dataset using (A) HierbaNetV1, and (B) InceptionV3.

Real-time inference

HierbaNetV1 is employed with HierbaApp (Michael & Manivasagam, 2024b), an Android mobile application that distinguishes sorghum crops from its associated weeds. We have used four different equipments of varying types and resolutions to capture and detect real-time crops and weeds using HierbaApp. Field images captured using Canon 80 D, Canon 600 D, Nixon CoolPix, and Samsung Galaxy M31 are tested in HierbaApp, which employs HierbaNetV1 for prediction at its backend. Test results with their respective image, ground truth, and prediction are depicted in Fig. 11. Real-time inference using HierbaApp with the application’s live prediction results are illustrated in Fig. 12. We tested 16 real-time images among which 15 are true positives and one is false negative. As HierbaNetV1 predicts research objects with high true positives irrespective of the equipment used, we state that our novel architecture is generalized and is suitable for weed detection in real-world agricultural settings. Nowadays Android devices are most commonly used among farmers and hence we consider mobile-based crop-weed detection is a good choice for manual detection with a minimum computational requirement and low-cost deployment.

Figure 11 Real-time in-field images using four equipments for detection: (A–C) using Canon 80 D; (D–F) using Canon 600 D; (G–I) using Nixon CoolPix; and (J–l) using Samsung Galaxy M31 along with their respective ground truth and prediction.

Figure 12 Real-time inference using HierbaApp, which employs our novel architecture ‘HierbaNetV1’ for crop-weed detection.

(A–L) The detection results of the respective images in Fig. 11.

Comparative analysis with pre-trained and SOTA models

Besides HierbaNetV1, five pre-trained architectures namely, InceptionV3, ResNet152V2, VGG19, DenseNet201, and MobileNetV2 are also trained, validated, and tested on the “SorghumWeedDataset_Classification”. Among the pre-trained models, the highest accuracy of 0.9791 is produced by InceptionV3, and the lowest loss of 0.1444 by VGG19. Confusion matrices on HierbaNetV1 and InceptionV3 are depicted in Figs. 13B and 13D for comparison. The test analysis is further extended to perform a thorough examination using other metrics such as precision, recall, and F1-score which are shown in Table 5. Results once again prove our proposed architecture surpasses other pre-trained models.

Figure 13 Confusion matrix of SorghumWeedDataset_Classification with: (A) HierbaNetV1 one block, (B) HierbaNetV1 two blocks, (C) HierbaNetV1 three blocks, and (D) InceptionV3.

Table 5 Performance evaluation of HierbaNetV1 against pre-trained models on SorghumWeedDataset_Classification using accuracy, precision, recall, F1-score, and loss.

Model	Accuracya	Precision	Recall	F1-score	Loss	
HierbaNetV1 (Proposed model)	0.9861	0.9860	0.9862	0.9860	0.0700	
Pre-trained models	
InceptionV3	0.9791	0.9795	0.9795	0.9792	1.5472	
VGG19	0.9698	0.9704	0.9702	0.9698	0.1444	
ResNet152V2	0.9675	0.9685	0.9681	0.9676	1.4649	
DenseNet201	0.9582	0.9601	0.9590	0.9583	1.0096	
MobileNetV2	0.9327	0.9336	0.9321	0.9323	2.2053	
SOTA architectures	
Hybrid CNN-Transformer	0.9606	0.9604	0.9605	0.9605	1.1503	
DarkNet53	0.9397	0.9395	0.9397	0.9395	2.5037	
RCNN	0.9142	0.9140	0.9142	0.9140	3.0356	
Note:

a The table is organized based on accuracy within each section.

To definitively prove HierbaNetV1’s advantage over SOTA techniques, the experimental findings are contrasted with methods of varying capacities and architectures. Three SOTA weed classification methods such as RCNN, DarkNet53, and Hybrid CNN-Transformer are evaluated under the same conditions as HierbaNetV1. The results conclusively demonstrate the superiority of HierbaNetV1 over SOTA approaches and are also given in Table 5.

Ablation study

The impact of each BLOCK in HierbaNetV1 is investigated through an ablation study utilizing the following architecture variations. To determine the best and most effective model, the three variants are trained, validated, and tested on SorghumWeedDataset_Classification. Figures 13A–13C portray the confusion matrices of HierbaNetV1 with one block, two blocks, and three blocks respectively. This shows the architecture with two blocks yields the best performance in contradiction with one block and three blocks. Additionally, we have examined the feature maps generated at different layers to understand the details of features in each block. Feature maps illustrated in Fig. 14 make abundantly evident that block 1 and block 2 generate rich information with low-level and high-level features which enables effective learning. In contrast, block 3 has a low spatial dimension with high information loss and is not taken into consideration. We can hereby conclude that HierbaNetV1 performs better when two-block feature extraction is used.

Figure 14 Ablation study conducted on HierbaNetV1 with three blocks to comprehend each block’s contribution: feature maps generated by (A) Layer 1-Base_Conv1 (Conv2D), (B) Layer 25-B1_LL_Conv1 (Conv2D), (C) Layer 33-Base_Conv2 (Conv2D), (D) Layer 57-B2_LL_Conv1 (Conv2D), (E) Layer 65-Base_Conv3 (Conv2D), (F) Layer 89-B3_LL_Conv1 (Conv2D), and (G) Layer 97-Base_Conv4 (Conv2D).

Component analysis

One input layer, 19 convolutional layers, 19 batch-normalization layers, 19 activation layers, seven dimensionality reduction layers, four feature integration layers, one global average pooling layer, one dropout layer, and one dense layer make up the total of 72 layers in HierbaNetV1.

With a significantly lower parametric complexity of 14.3 M, the model is less complex due to the significantly lower number of convolutional layers. The architecture is designed carefully while increasing the depth and width of the network parallelly thus avoiding a very deep neural network. The 14 convolutional layers in the series ‘B1_HL_ConvX(Conv2D)’ and ‘B2_HL_ConvX(Conv2D)’ utilizes four diversified filters with kernel sizes 1×1, 3×3, 5×5, and 7×7 thus creating a rich content feature irrespective of the ROI size. B1_LL_Conv1(Conv2D) and B2_LL_Conv1(Conv2D) propagate the low-level features thus enabling basic features of crops and weeds. This propagation also avoids the vanishing gradient problem.

The 19 batch normalization layers succeed the 19 convolutional layers to provide a firm convergence and reduce the necessity for other regularization techniques in the architecture. The LeakyReLU activation function addresses the dying ReLU problem in the proposed method (Dubey & Jain, 2019). Drop-out acts as a model regularizer and thus solves the problem of overfitting.

In addition, early stopping helps to find the perfect balance between the bias and variance, thus solving the ‘Bias-Variance trade-off’ and avoiding over-fitting in the proposed model (Yang et al., 2020). The implementation of stratified 10-fold cross-validation is used for model training and validation to allow model generalization for unknown test data. Furthermore, the Glorot Uniform weight initializer solves the problem of vanishing gradients and exploding gradients, thus helping HierbaNetV1 to converge faster.

Conclusions and future enhancements

Dataset scarcity is a big challenge in today’s research community. Consequently, This research contributes an open-access crop-weed classification dataset termed SorghumWeedDataset_Classification to solve the crop-weed classification problem using Computer Vision techniques. This research also contributes a novel feature extraction framework code-named HierbaNetV1 for intensive and diversified feature extraction. Furthermore, encouraging weed research the trained weights, and the Python implementation of HierbaNetV1 are made publicly available. Subsequently, HierbaNetV1 can be used as a pre-trained architecture to address the AI-based classification problems of all categories. HierbaNetV1 gives an overall Top-1 testing accuracy of 0.986 for SorghumWeedDataset_Classification. After performing extensive experiments on our proposed architecture, HierbaNetV1 proved to be an effective feature extractor.

HierbaNetV1 requires devices with powerful GPUs and high RAM, which is not available easily. Training and validating the model with such system specifications are expensive. Consequently, in the near future, an enhanced version of the proposed architecture which is lightweight and utilizes fewer resources will be built. Furthermore, HierbaNetV1 is currently expanding as “HierbaNetV1_MRCNN” that locates smaller ROIs precisely in weed detection, localization, and segmentation tasks. The HierbaNetV1_FPN feature pyramid network (FPN), which is built on HierbaNetV1, is enforced in this extended study to locate the ROIs. Furthermore, HierbaApp will be enhanced to support agriculturalists. Steps are taken to design HierbaRobo-a smart weed-removing robot for small-scale agricultural lands. This work is also left open to the research community and appreciate novel ideas.

The authors thank the SRM Institute of Science and Technology, Kattankulathur Campus, Tamil Nadu, India for providing the land for data acquisition and continuous support during land preparation, sowing, weeding, irrigation, and all other aspects till harvest.

The authors also thank Dr. M. Jawaharlal, Dean, SRM College of Agricultural Sciences, SRM Institute of Science and Technology, Baburayanpettai, Maduranthagam (TK), Chengalpattu (Dt), Tamil Nadu, India, and Dr. S. Marimuthu, Agronomist, Assistant Professor, and Head of the Department of Crop Management at SRM College of Agricultural Sciences for their support in specie selection, planting patterns, segregating data samples to respective classes, ground truth preparation through annotation and guidance to understand the in-depth about crops and weeds.

The authors thank Dr. Revathi Venkataraman, Professor and Chair Person, School of Computing, and Dr. Annapurani Panaiyappan K, Professor and Head, Department of Networking and Communications, SRM Institute of Science and Technology, Kattankulathur, Tamil Nadu, India for providing access to the computing resources for training, validating, and testing our findings and to carry forward our research.

Additional Information and Declarations

Competing Interests

Author Contributions

Patent Disclosures

Data Availability

The authors declare that they have no competing interests.

Justina Michael conceived and designed the experiments, performed the experiments, analyzed the data, performed the computation work, prepared figures and/or tables, authored or reviewed drafts of the article, and approved the final draft.

Thenmozhi Manivasagam conceived and designed the experiments, performed the experiments, analyzed the data, performed the computation work, prepared figures and/or tables, authored or reviewed drafts of the article, and approved the final draft.

The following patent dependencies were disclosed by the authors:

Patent application no: 202441050194

Date: 01.07.2024

Title: AN INTENSIVE FEATURE EXTRACTION FRAMEWORK.

The following information was supplied regarding data availability:

The raw data is available at Mendeley Data: Michael, Justina; M, Thenmozhi (2023), “SorghumWeedDataset_Classification”, Mendeley Data, V1, doi: 10.17632/4gkcyxjyss.1.

The code is available at Code Ocean: Justina Michael J, Thenmozhi M (2024) Implementation Of HierbaNetV1-A Novel Convolutional Neural Network Architecture [Source Code]. https://doi.org/10.24433/CO.1923872.v1.

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
