# Peer review of "HierbaNetV1: a novel feature extraction framework for deep learning-based weed identification"

_PeerJ Computer Science, doi:10.7717/peerj-cs.2518_

## Round 0.1 · original submission · Major Revisions

Thank you for submitting your manuscript to PeerJ Computer Science. The review process has been completed, and we have carefully considered the feedback provided by the reviewers.

The reviewers have acknowledged the potential value of your work but have raised several significant concerns, particularly regarding the methodology and experimental evaluation. These concerns require substantial revisions to ensure that the manuscript meets the rigorous standards of our journal.

In light of these comments, I am recommending that your manuscript undergoes a major revision. We encourage you to carefully address each of the reviewers’ comments, paying close attention to the methodological issues and the robustness of your experimental evaluation. A detailed response to the reviewers, explaining the changes made or providing justifications for any unaddressed points, should accompany your revised submission.

Once the revisions have been completed, your manuscript will undergo a further round of review to ensure that all major concerns have been satisfactorily addressed.

We appreciate the effort that you have put into this research and look forward to receiving your revised manuscript.

Reviewer 1 ·

Basic reporting

The yield loss due to weed must be reported. A review of the literature is lacking and that can be improved by including other research papers on various crops such as sugarcane from the same country.

Experimental design

Data set details are lacking, how was it acquired like lighting conditions and other environmental parameters?

Validity of the findings

Pl validate with manual detection

Reviewer 2 ·

Basic reporting

This manuscript presents a well-organized study that addresses the critical issue of crop and weed Classification in precision agriculture. The authors introduce HierbaNetV1, a Convolutional Neural Network architecture, and its associated dataset, SorghumWeedDataset Classification.

Evaluating the model's performance with more diverse datasets: While the performance on the SorghumWeedDataset Classification is noteworthy, assessing its capability on a more diverse set of datasets could further validate its generalizability and robustness across different crop types and environmental conditions.

Experimental design

Detailed ablation study: Although an ablation study is mentioned, providing more detailed insights into how various components of HierbaNetV1 contribute to its overall performance would be beneficial, which would help understand each module's importance and potentially guide further optimizations.

Validity of the findings

Real-world application: The manuscript would benefit from discussing the practical implementation of HierbaNetV1 in real-world agricultural settings, which includes considerations of computational requirements, deployment feasibility, and potential challenges in field conditions.

Additional comments

I encourage the authors to update the literature by discussing the pros and cons of recent related papers.

Reviewer 3 ·

Basic reporting

- While this study has the potential to contribute to the field, the authors have not sufficiently articulated the urgency or significance of the research. The background provided lacks depth, making it difficult to fully appreciate the study's relevance.

- The paper’s structure currently resembles a report rather than a well-organized academic paper. For instance, in Section 2, the authors focus primarily on describing well-known methods. This section would benefit from a more critical discussion of the research gap and the specific problems that the proposed method addresses. It is crucial for the authors to clearly differentiate their approach from existing methods to highlight its unique contributions.

Experimental design

- While the proposed method shows promise, the current experimental evaluation is insufficient to conclusively demonstrate its superiority over state-of-the-art approaches. The comparison lacks recent methods, particularly those based on transformer or hybrid architectures. I recommend including comparisons with methods of varying capacities and architectures to provide a more comprehensive evaluation.

- Evaluating the proposed method solely on a classification task provides a general context, but it does not fully showcase its potential. To strengthen the validation of the method, it would be valuable to include experiments on downstream tasks such as semantic segmentation or instance segmentation. This would better illustrate the effectiveness of the proposed method as a backbone for more complex tasks.

Validity of the findings

- There is no discussion of the experimental results.

---

## Round 0.2 · Minor Revisions

Thank you for submitting your manuscript to PeerJ Computer Science. After careful review, the reviewers have raised some concerns regarding the methodology and experimentation that need to be addressed before we can proceed with the publication.

We kindly request that you revise your manuscript in light of the reviewers' comments and make the necessary adjustments. Please also provide a detailed response letter addressing each of the reviewers' suggestions and observations.

We are confident that, with these revisions, your manuscript will be considered for publication.

Thank you again for your contribution, and we look forward to receiving your revised submission.

Reviewer 1 ·

Basic reporting

Satisfactory included what was suggested in the first review.

Experimental design

Well defined.

Validity of the findings

Validated with manual method.

Additional comments

NA

Reviewer 2 ·

Basic reporting

The authors answered my questions and revised the manuscript according to my comments.

Experimental design

The experimental design is correct..

Validity of the findings

The obtained results are good.

Additional comments

The revised manuscript is acceptable.

Reviewer 3 ·

Basic reporting

The writing style of this paper still resembles a report rather than a formal academic research article. I strongly recommend the authors consult recently published papers from this journal and adhere to the prescribed guidelines. Furthermore, the manuscript contains an excessive number of unnecessary subsections, detracting from its coherence.

Experimental design

Table 1 requires further clarification. The authors merely list a collection of papers without providing any substantive analysis. These studies address distinct tasks and utilize different datasets, yet no attempt has been made to critically assess or synthesize the information presented.

Validity of the findings

The comparison provided in Section 5.3 is problematic. The authors directly compare their method with the reported performance of other studies, despite these works using different datasets and experimental setups, as evidenced in Table 1. Such a comparison is not valid and should be conducted under identical conditions. Moreover, it is unclear why this comparison is discussed in a separate section. Why not incorporate it into Table 5 for a more streamlined presentation?

Additional comments

I commend the authors for their efforts in addressing the reviewers' comments. While some responses are adequate, several key points remain insufficiently addressed.

---

## Round 0.3 · accepted · Accept

I hope this message finds you well. After carefully reviewing the revisions you have made in response to the reviewers' comments, I am pleased to inform you that your manuscript has been accepted for publication in PeerJ Computer Science.

Your efforts to address the reviewers’ suggestions have significantly improved the quality and clarity of the manuscript. The changes you implemented have successfully resolved the concerns raised, and the content now meets the high standards of the journal.

Thank you for your commitment to enhancing the paper. I look forward to seeing the final published version.